# Temporal genome-wide fitness analysis of *Mycobacterium marinum* during infection reveals the genetic requirement for virulence and survival in amoebae and microglial cells

Louise H. Lefrançois,[1] Jahn Nitschke,[1] Huihai Wu,[2] Gaël Panis,[3] Julien Prados,[3,4] Rachel E. Butler,[2] Tom A. Mendum,[2] Nabil Hanna,[1] Graham R. Stewart,[2] Thierry Soldati[1]

**ABSTRACT** Tuberculosis remains the most pervasive infectious disease and the recent emergence of drug-resistant strains emphasizes the need for more efficient drug treatments. A key feature of pathogenesis, conserved between the human pathogen *Mycobacterium tuberculosis* and the model pathogen *Mycobacterium marinum,* is the metabolic switch to lipid catabolism and altered expression of virulence genes at different stages of infection. This study aims to identify genes involved in sustaining viable intracellular infection. We applied transposon sequencing (Tn-Seq) to *M. marinum*, an unbiased genome-wide strategy combining saturation insertional mutagenesis and high-throughput sequencing. This approach allowed us to identify the localization and relative abundance of insertions in pools of transposon mutants. Gene essentiality and fitness cost of mutations were quantitatively compared between *in vitro* growth and different stages of infection in two evolutionary distinct phagocytes, the amoeba *Dictyostelium discoideum* and the murine BV2 microglial cells. In the *M. marinum* genome, 57% of TA sites were disrupted and 568 genes (10.2%) were essential, which is comparable to previous Tn-Seq studies on *M. tuberculosis* and *M. bovis*. Major pathways involved in the survival of *M. marinum* during infection of *D. discoideum* are related to DNA damage repair, lipid and vitamin metabolism, the type VII secretion system (T7SS) ESX-1, and the Mce1 lipid transport system. These pathways, except Mce1 and some glycolytic enzymes, were similarly affected in BV2 cells. These differences suggest subtly distinct nutrient availability or requirement in different host cells despite the known predominant use of lipids in both amoeba and microglial cells.

**IMPORTANCE** The emergence of biochemically and genetically tractable host model organisms for infection studies holds the promise to accelerate the pace of discoveries related to the evolution of innate immunity and the dissection of conserved mechanisms of cell-autonomous defenses. Here, we have used the genetically and biochemically tractable infection model system *Dictyostelium discoideum*/*Mycobacterium marinum* to apply a genome-wide transposon-sequencing experimental strategy to reveal comprehensively which mutations confer a fitness advantage or disadvantage during infection and compare these to a similar experiment performed using the murine microglial BV2 cells as host for *M. marinum* to identify conservation of virulence pathways between hosts.

**KEYWORDS** *Mycobacterium marinum*, phenotypic profiling, *Dictyostelium discoideum*, BV2 microglial cells, essentiality, Tn-Seq

M ycobacterium marinum is the causative agent of a tuberculosis-like disease in poikilotherms and is also an opportunistic human pathogen, where the infection

Address correspondence to Thierry Soldati, thierry.soldati@unige.ch, or Nabil Hanna, Nabil.hanna@unige.ch.

Louise H. Lefrançois and Jahn Nitschke contributed equally to this article. They are listed both alphabetically and in order of decreasing seniority.

The authors declare no conflict of interest.

See the funding table on p. 23.

is limited to the skin and extremities due to growth restriction at about 30°C (1). In contrast to *Mycobacterium tuberculosis,* which has evolved as a strict human pathogen, *M. marinum* remains a generalist and has the capacity to affect a wide range of animal hosts and protozoa. Although *M. marinum* has a larger genome than *M. tuberculosis*, they are phylogenetically closely related and the processes accompanying infection at a cellular level are similar (2, 3).

Mycobacteria possess a remarkably elaborate cell wall to protect themselves against environmental stresses and chemicals (4, 5), and components of the cell wall are well conserved between *M. tuberculosis* and *M. marinum*. The most remarkable is a specific hydrophobic outer layer, called the "mycomembrane," composed of several lipids including mycolic acids (MA) and the complex and branched phthiocerol dimycocerosates (PDIMs) (6, 7). The mycobacterial cell wall can be considered a "lipidic virulence factor" and is essential to establish the infection and also to persist in a non-replicative state during dormancy. The stages of infection of *M. tuberculosis* and *M. marinum* in a variety of evolutionary distant phagocytes are extremely well conserved, and a single cycle of infection lasts about 48 hours (illustrated in Fig. 1A). After entry by phagocytosis in animal phagocytes of the innate immune system (8) or in amoebae (3, 9), secretion of virulence factors, and especially of EsxA, via the ESX-1 secretion system inflicts membrane damage to the Mycobacteria-containing Vacuole (MCV) (10–12), which are repaired by a combination of cytosolic machinery. After cycles of damage and repair, the mycobacteria reach the cytosol where they continue to proliferate until they disseminate via lytic and non-lytic mechanisms (13–16).

Intracellular mycobacteria use host lipids as a carbon and energy source during active growth and dormancy (17, 18). *M. tuberculosis*, in particular, has the unique ability to use cholesterol and fatty acids as sole carbon sources *in vivo* and *in vitro* (19). It has been demonstrated that *M. marinum* can use fatty acids derived from triacylglycerols in lipid droplets as well as from host membrane phospholipids for its metabolism and storage (20). To access the host lipids required for its survival, intracellular mycobacteria need specific transporters such as those of the MCE family that are conserved among many bacteria species including mycobacteria. Of the four *mce* operons present in the genome of mycobacteria (*mce1-4*), two have been characterized *in vitro* and *in vivo* as important for the uptake of cholesterol (*mce4*) and fatty acids (*mce1*) (21).

The social amoeba, *Dictyostelium discoideum* is an alternative phagocytic host model to study interactions with environmental and pathogenic bacteria (9, 22). The phagocytic pathway is highly conserved and represents a major restriction point for bacteria in both macrophages and amoebae. Nevertheless, *M. marinum* can survive inside *D. discoideum* in a similar way as *M. tuberculosis* in macrophages (Fig. 1A) (3, 22, 23). *D. discoideum* has emerged as a powerful genetically and biochemically tractable model to study processes of *M. marinum* infection, and in particular has recently been used to demonstrate the role of the ESCRT and autophagy machinery in MCV membrane repair (24, 25), the role of metal poisoning in bacterial growth restriction (26, 27) as well as to identify anti-infective compounds (28, 29). Therefore, we use *D. discoideum* and *M. marinum* as a model system to study cell-autonomous defense mechanisms that are relevant to the pathogenesis of tuberculosis (3, 8, 9).

Although the *D. discoideum* host system has been exploited mainly with the use of targeted, candidate-based approaches, to investigate interactions with pathogenic bacteria (23, 30), it is also highly suitable to more systematic approaches such as proteomic profiling of the MCV (31), or transcriptomics survey of pathogen and host by dual RNA-Seq (32).

Transposon mutagenesis sequencing (Tn-Seq) is a method used to identify essential or relevant genes in bacterial survival or growth under specific conditions or selections. The principle is based on identifying the location and the frequency of the transposon insertions in a library of mutants by sequencing the transposon-genome insertion sites. The Mariner transposon, used in the present study, inserts at every TA dinucleotide sequence throughout the genome. The depth of sequencing allows the quantification of

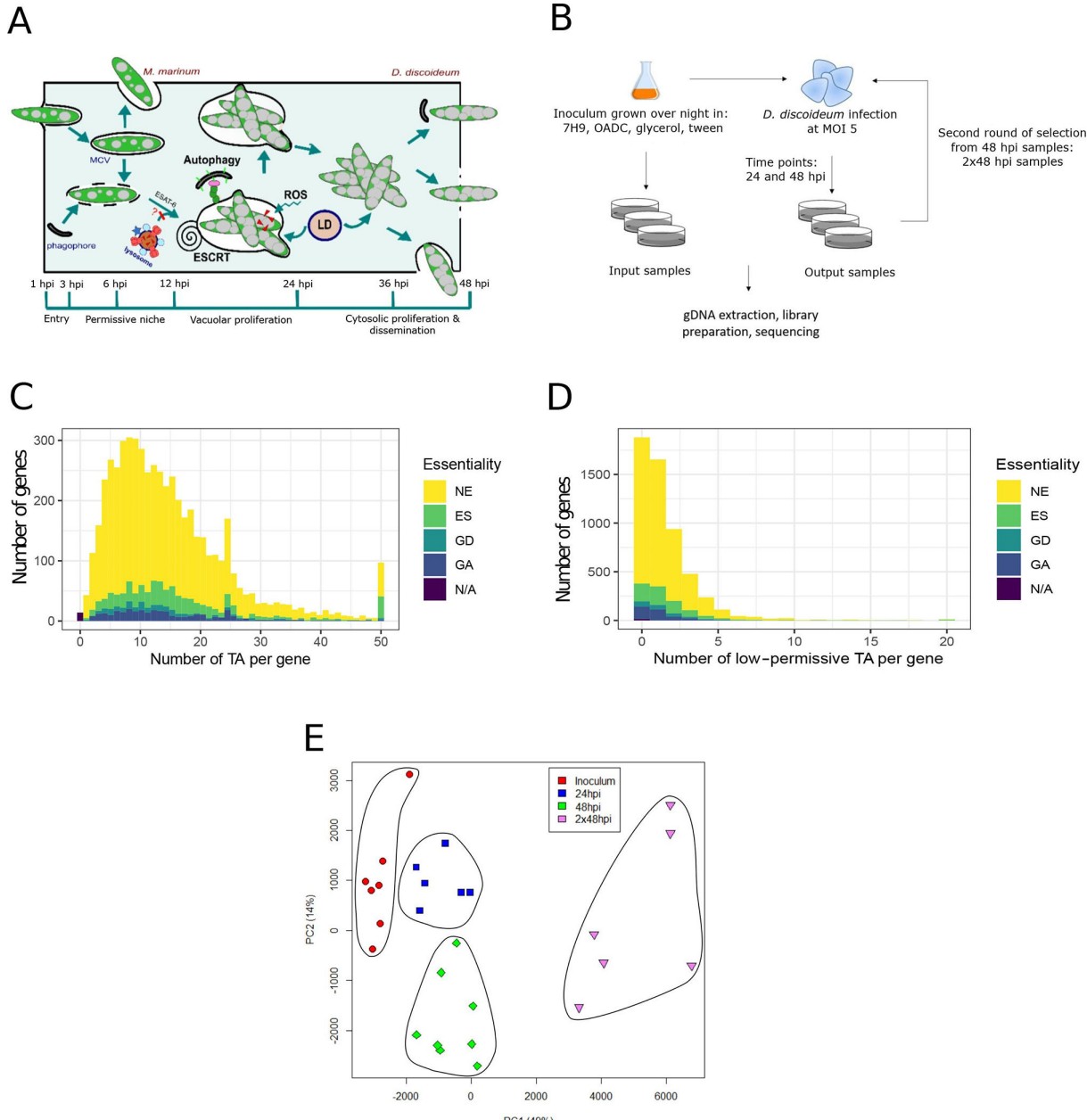

**FIG 1** Description of the selection, library coverage, and sample selection. (A) *M. marinum* infection course in *D. discoideum* over 48 hours. *M. marinum* is phagocytosed by *D. discoideum* and rapidly manipulates its phagocytic pathway by blocking the phagolysosomal fusion to reside within a replicative niche (MCV). *M. marinum* proliferates within its MCV which finally breaks and releases mycobacteria to the cytosol where the bacteria continue to grow. (B) Workflow of *M. marinum* library selection during infection in *D. discoideum*. *M marinum* stably expressing m-Cherry was thawed and grown overnight. A sample was collected which constituted the "input sample" or inoculum, the remaining culture was used to infect *D. discoideum* AX2 (Ka) and samples were taken over a time course of 24 and 48 hpi. *M. marinum* was recollected from the 48 hpi samples and used to infect again *D. discoideum*. After another 48 hpi, samples were collected which, for simplicity, are called 2 × 48 hpi samples. (C) Histogram generated from the Hidden Markov Model (HMM) analysis using TRANSIT, representing the essentiality of the 27 samples of *M. marinum* mutant libraries. The calls for different regions are divided into five categories: (1) essential regions (ES) which are mostly devoid of insertions (2), non-essential genes (NE) containing read-counts around the global mean (3), growth-defect genes (GD), and (4) growth-advantage genes (GA) which represent genes with significantly increased or decreased normalized read counts, respectively (5), genes without any TA site. (D) Identification of low-permissive TA sites in the *M. marinum* genome was performed for each of the 102,057 TA sites. We determined the surrounding ±3 bp sequence and checked whether it matches with the pattern "SGNTANCS" which has been reported to be "low-permissive" to transposon insertions. The histogram reports gene essentiality in dependence on the number of low-permissive TA sites. The same categories as for panel C were applied here. (E) Principal components 1 and 2 of normalized insertion counts, colored by conditions: Inoculum (*n* = 7), 24 hpi (*n* = 6), 48 hpi (*n* = 8), and 2 × 48 hpi (*n* = 6). Below the axis label, the percentage of variance captured by the respective principal component is denoted.

the relative frequency of individual mutants in the pool before and after selection (33). The Tn-Seq method has been performed with many environmental and pathogenic bacteria (34–37), including *M. tuberculosis* (38–41), before being adapted to other mycobacteria (42–44). To date, only one Tn-Seq study performed in *M. marinum* using the E11 strain has been published (45). However, comparative genome analysis of *M. marinum* strains isolated from diseased fish or human patients suggests that the E11 strain and the M strain likely belong to two different clusters (46) and might have different host ranges. For example, *M. marinum* M is virulent for all the model hosts studied so far, from amoebae to flies, fish, and mice (47, 48). On the contrary, the E11 strain is avirulent for *D. discoideum* (45).

While several screens have been employed to identify genes that are important for mycobacteria survival *in vitro* and *in vivo*, in the present study, we used the *M. marinum* strain M, isolated from a human patient, and commonly used as a model for *M. tuberculosis* pathogenesis (3, 8, 22). We identified essential genes and performed a comparison with previously published sets of essential genes from *M. marinum* strain E11, *M. tuberculosis,* and *M. bovis*. We then examined the fitness impact of genome-wide insertion mutations at various stages of infection in both phagocytes, the amoeba *D. discoideum* and murine BV2 microglial cells, providing invaluable insights into the temporal genetic requirements for *M. marinum* to establish an infection.

## RESULTS AND DISCUSSION

### Selection process and distribution of the transposon insertions

The *M. marinum* strain M (hereafter described as *M. marinum*) library was generated with the phage MycoMarT7 vector carrying a *Himar1* transposon that inserts at the majority of TA dinucleotides in the genome, as previously described for *M. bovis* and *M. tuberculosis* (41, 42). For each infection, a frozen aliquot of the *M. marinum* library was grown in 7H9 to generate the reference input sample (hereafter called "Inoculum"), and the transposon insertion sites were sequenced. The selection was performed in two consecutive rounds of *D. discoideum* infection. As illustrated in Fig. 1A, a full infection cycle, from uptake, to MCV genesis and breakage, to dissemination takes 48 hours (3). To sample the early and late stages of a 48-hour infection cycle (Fig. 1A), *M. marinum* was collected from populations of infected *D. discoideum* at 24 hpi and 48 hpi. To increase the impact of the selection, we performed an iteration of a full infection cycle. For this, the 48 hpi sample was grown in 7H9 overnight and used as an inoculum (hereafter called "Inoculum_bis") for a second round of infection resulting in the "2 × 48 hpi" condition (Fig. 1B).

The genome of *M. marinum* has a length of 6,660,144 bp with one chromosome of 6,636,827 bp and one plasmid of 23,317 bp (2). The genome features 102,057 TA sites, theoretically allowing for an insertion every 65 bp on average, with 82.6% (84,295 sites) found within annotated genes and 17.4% (17,762 sites) in intergenic regions. Based on the sequencing of seven inoculum samples, we observed insertion in 57% (58,137 sites) of the TA sites (criteria: average normalized read count ≥1) (Fig. 1C; Table S1 at https://tinyurl.com/5dbxww5w). Recently, the presence of low-permissive sequences with consensus "(GC)GNTANC(GC)" has been highlighted in the *M. tuberculosis* genome (49). In the *M. marinum* genome, we found 9,370 TA sites (9.2% of the total) embedded in a low-permissive sequence. In agreement with that prediction, only 711 (7.6%) of these sites had an insertion. We then investigated the impact of these low-permissive TA sites on the gene essentiality status identified by TRANSIT (Fig. 1D; Table S1 at https://tinyurl.com/5dbxww5w). To retain a maximum of information, a combination of the two available annotations for the genome of *M. marinum* strain M (NC_010612.1 and CP000854.1) was used in addition to the plasmid found in the M strain pMM23 (NC_010604.1), resulting in a total number of 5,480 annotated genes. Because our primary aim was to measure the fitness impact of mutations, we concentrated on insertions that disrupt coding sequences, as they are expected to lead to loss of function. In addition, this approach allows us to directly compare our results with previously reported data on *M. marinum* strain E11, *M. tuberculosis,* and *M. bovis*, where authors

also focused on insertions in coding regions. All the TA sites are listed in Table S1 (https://tinyurl.com/5dbxww5w) and those mapping to coding sequences are shown in File S1 (https://tinyurl.com/5dbxww5w).

We found that, among the 651 essential genes identified in the inoculum with MMAR codes (Fig. S2), 56 of them have more than 25% low-permissive TA sites, raising the possibility of false positives. Within the 568 genes comprised in the essential core (see definition below), 192 genes exhibit a low-permissive TA site percentage of ≤5%, while six genes have a percentage of ≥50%. This indicates that our collection of core essential genes remains relatively unaffected by low-permissive TA sites (Table S2). Please note that the 84,495 sites mapping to genes include 84,295 unique TA sites and 200 TA sites that map to overlapping gene predictions (https://www.ncbi.nlm.nih.gov/genome/annotation_prok/) and are thus counted twice in the gene-wise summary in Table S2 compared to Table S1. Principal component analysis (PCA) of all 33 samples from each pool (Inoculum $n = 7$; 24 hpi $n = 6$; 48 hpi $n = 8$; Inoculum_bis $n = 6$; and 2 × 48 hpi $n = 6$) revealed excellent grouping by condition, except for the 48 hpi and Inoculum_bis samples, which show a major overlap (Fig. S1A). Removal of the Inoculum_bis samples did not impact significantly the PCA analysis of the other samples (Fig. 1E), and therefore this sample group was excluded from further TnSeq analysis. The remaining 27 samples were included in the subsequent analyses. In addition, we estimate that the additional power raised by comparing all conditions at once instead of having to integrate two separate comparisons (each to its inoculum) far outweighs the small potential bias introduced.

## Identification of the core of essential genes and comparison to other mycobacteria

The data were analyzed using TRANSIT, which is software that automates the analysis of Himar1 Tn-Seq data sets (50). TRANSIT uses a Bayesian method and a hidden Markov model (HMM) which is implemented as a 4-state model assigning the genes into four categories: essential regions (ES) which are mostly devoid of insertions, non-essential genes (NE) containing read-counts around the global mean, growth-defect genes (GD), and growth-advantage genes (GA) which represent genes with significantly increased or decreased normalized read-counts, respectively.

TRANSIT identified 651, 635, 630, and 704 essential genes in the inoculum, 24 hpi, 48 hpi, and 2 × 48 hpi groups, respectively. In all, 568 genes (10.2%) were essential in all conditions and this intersection was deemed to be the "essential core" (Fig. 2A; Table S2). In principle, insertion mutants in genes that are essential for growth in non-selective broth (a total of 651) should be absent from the starting pool (the Inoculum) and, therefore, should also not appear at any later stage of any selection, including after infection of a phagocyte. Indeed, a few such mutants are detected, for example, 12 genes in the inoculum are essential but ceased to be defined as such during any stage of infection. We consider that the detection of a low number of essential genes not shared between all conditions (between 1 and 20) reflects the noise level of the strategy and analysis. By contrast, the 42 essential genes that are specific for the 2 × 48 hpi condition are discussed below. Therefore, we are convinced that our operational definition of the essential core greatly helps to reduce the number of false positives, thus allowing us to concentrate on truly essential genes.

Then, we compared the essential core of *M. marinum* to the sets of essential genes previously identified for other mycobacteria species, the *M. marinum* strain E11 (45), the *M. tuberculosis* H37Rv (40), and *M. bovis* (42) (Fig. 2B). First, we identified *M. marinum* strain M orthologs of ES genes reported in the three studies, and then we determined the intersection between these genes and our essential core. This interspecies intersection of essential genes was 37.3% ($N = 237$) with *M. marinum* E11 (45), 51.4% ($N = 311$) with *M. tuberculosis* H37Rv (40), and 27.7% ($N = 161$) with *M. bovis* (42). Furthermore, 96 orthologous genes were common between the "essential core" identified here and the three studies on *M. marinum* E11, *M. tuberculosis* H37Rv, and *M. bovis*. This

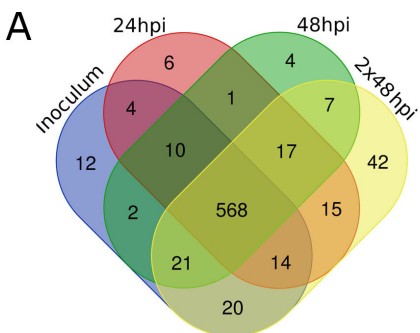

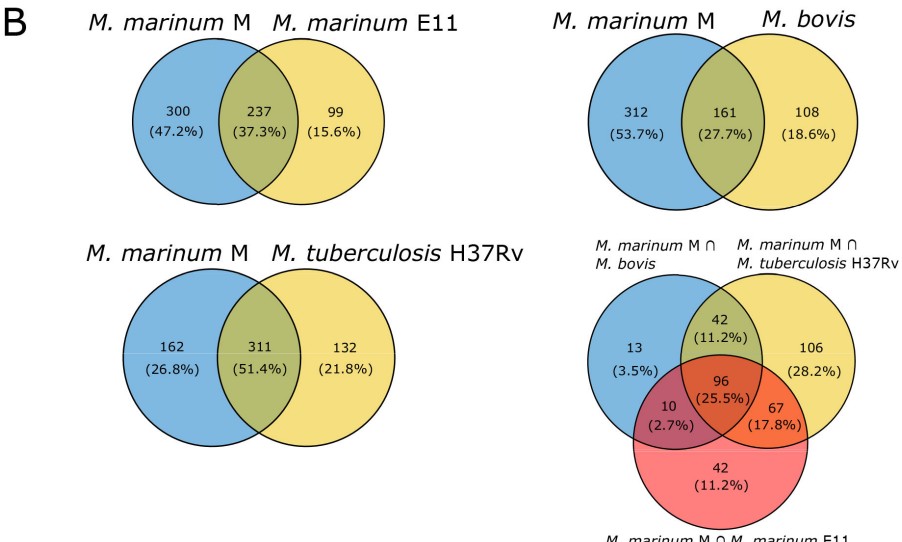

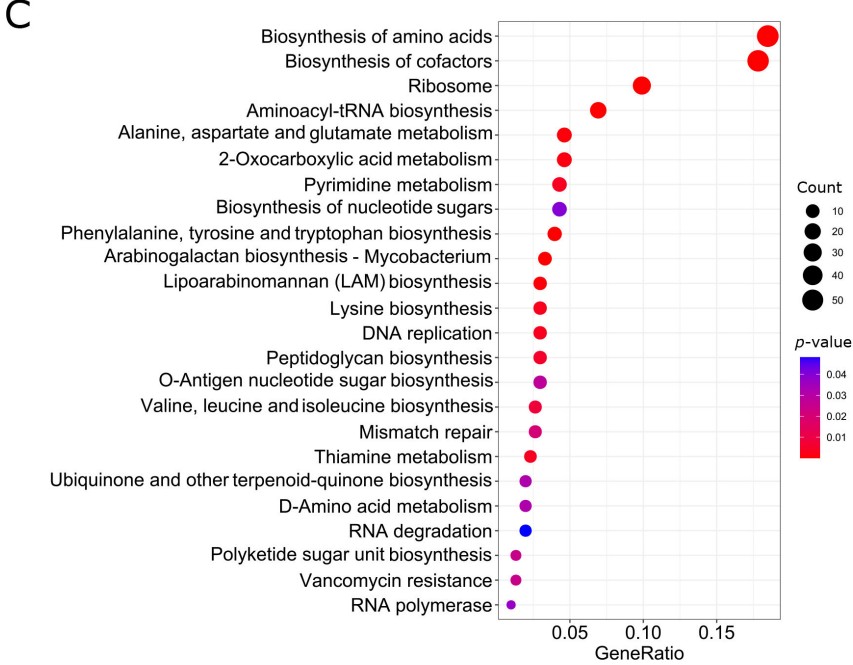

**FIG 2** Identification of the "essential core" of *M. marinum* in *D. discoideum*. (A) Essential genes for each condition were determined with TRANSIT. The Venn diagram shows the intersection of all four conditions, that is, the "essential core" containing 568 genes. (B) Comparison of the "essential core" of *M. marinum* M strain with the *M. marinum* E11 strain, the Mtb

**FIG 2** (Continued)

H37Rv strain, and the *M. bovis* strain. Comparisons were performed by considering only *M. marinum* genes from the MMAR nomenclature and with at least one orthologous gene in the compared strain. (C) KEGG terms enriched in the essential core are sorted as a dotplot. Using R and the package clusterProfiler terms were filtered by *P*-value < 0.05. The *P*-value is color coded, and the number of enriched genes found in the respective term is coded by dot size. GeneRatio equals the number of differentially expressed genes against the number of genes associated with a GO term in the *M. marinum* genome.

interspecies gene set includes genes involved in fundamental cellular processes such as DNA replication (GyrA, GyrB, DnaG) or ATP synthase genes, and nucleotide biosynthesis enzymes such as pyrimidine synthase and genes involved in the *de novo* purine synthesis (*purH)* (Fig. 2B; Table S2).

The *M. marinum* essential core was further characterized using a KEGG enrichment analysis to pinpoint overrepresented functional categories. Functional groups with a *P*-value < 0.05 are illustrated in a dotplot (Fig. 2C) and an enrichment map (Fig. S2). The size and color of the dots in the dotplot represented the number of genes in the KEGG term and their significance, respectively (Fig. 2C; Table S4). The major categories primarily encompassed intrinsic metabolic pathways essential for bacterial survival both *in vivo* and/or *in vitro*. Notably, the indispensability of DNA replication machinery was underscored, with significant enrichment of genes related to replication fork formation and movement (such as *dnaB*, encoding DNA helicase, *dnaG*, encoding DNA primase, and *dnaE1*, encoding DNA polymerase III). Genes involved in the repair of lagging strand Okazaki fragments (e.g., *polA*, encoding DNA polymerase I, and *ligA*, encoding DNA ligase) were also found to be essential.

Furthermore, essential functional groups encompassed genes related to cell wall synthesis, including those within the peptidoglycan biosynthesis pathway, such as *murB* and *ddlA*, encoding D-alanine D-alanine ligase. In addition, pathways for the biosynthesis of valine, leucine, isoleucine, and lysine were enriched, featuring genes like *ilvB1* (MMAR_1720) and *lysA*. These two genes are known to be essential in *M. tuberculosis* and mutations in these genes are only obtained when the medium is supplemented with the respective amino acids (51, 52). Importantly, a set of 42 essential genes specific to 2 × 48 hpi was identified. Among these, *icl1* (MMAR_0792), encoding the enzyme isocitrate lyase involved in the TCA cycle, was notable. Indeed, disruption of this gene has been linked to impaired chronic-phase persistence of *M. tuberculosis* in mice (53). In addition, *whiB1* (MMAR_1338), a transcriptional regulator belonging to the WhiB family, was found to be essential at 2 × 48 hpi. WhiB1 is known to play a role in cell growth and the initiation of dormancy in response to nitric oxide (NO) (54). Although *D. discoideum* does not produce NO to combat bacterial infections, WhiB1 might sense and respond to more general redox stress in this infection model. Furthermore, two genes involved in dihydrofolate biosynthesis *folB* (MMAR_5110) and *folP1* (MMAR_5111) were identified, with the latter being essential for *de novo* synthesis of folate. Inhibition of its enzymatic activity results in the depletion of the folate pool, leading to growth inhibition and bacterial death (55).

## Hierarchical clustering of time-series

We wanted to gain a deeper understanding of insertion frequencies resolved by time and infection stages. We calculated the fold change of Tn insertions per gene for each time point using the Inoculum as the reference. This analysis generated profiles for all mutated genes, and the complete list of profiles is available in Table S2 and File S1 (https://tinyurl.com/5dbxww5w). Then, we applied hierarchical clustering to the temporal profiles (see Material and Methods) and classified them into nine clusters with distinct patterns. The clusters contained between 387 (cluster 9) and 681 (cluster 1) genes (Fig. 3A). To delve deeper into the biological functions characterizing each cluster, we conducted a GO term enrichment analysis for each of them. Most clusters did not exhibit significantly enriched functional groups with the specified cutoff values (*P*-value ≤ 0.01) for this analysis, except for clusters 2, 7, and 9 which are represented

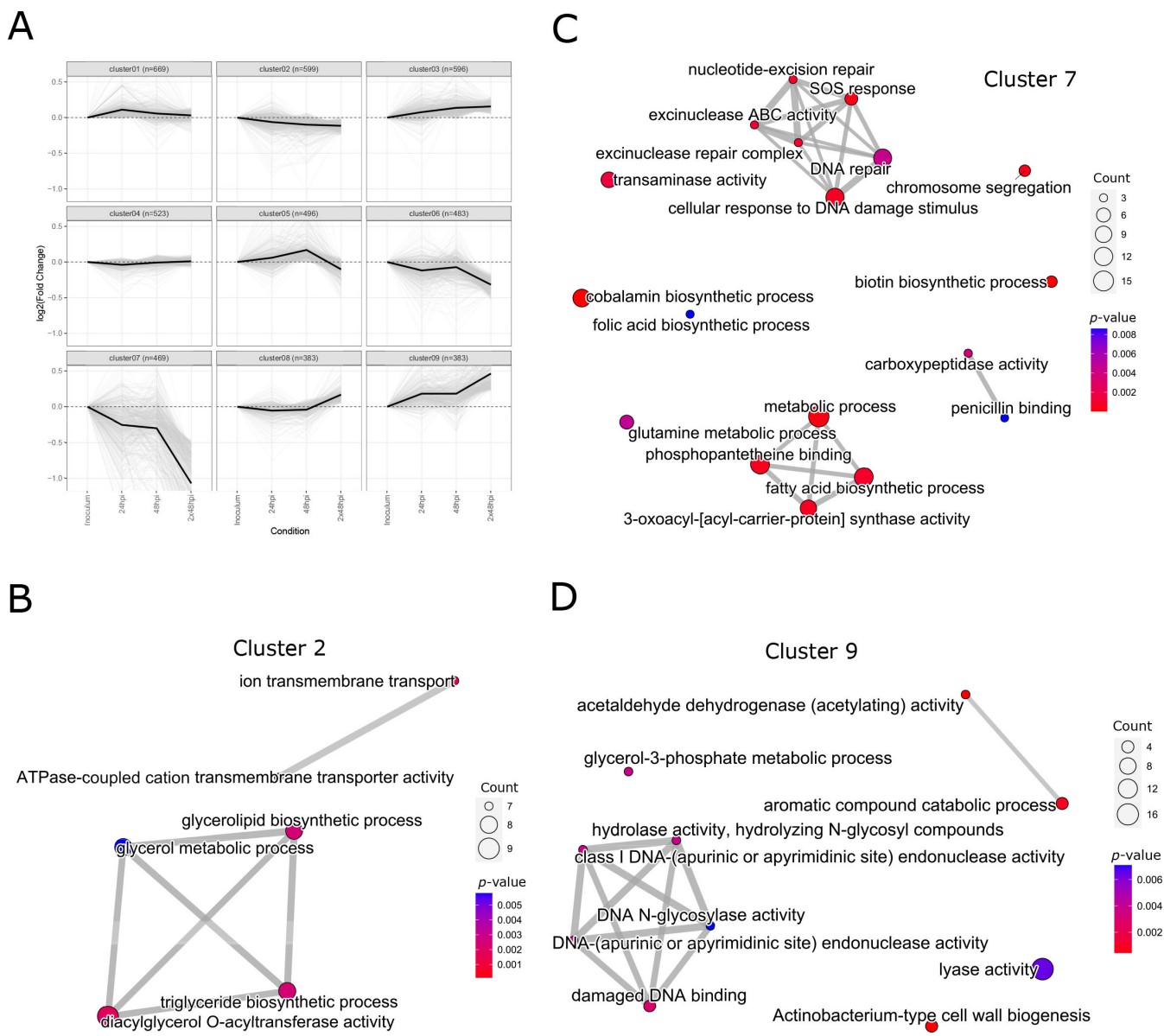

**FIG 3** Hierarchical clustering of log$_2$ fold changes in *D. discoideum*. The relative abundance of transposon insertions at various infection time points was compared to the initial inoculum. From these binary comparisons, a log$_2$fc and a *P*-value were computed. (A) Genes with at least 5 TA sites and a normalized read count >1 in any of the samples were selected. The Manhattan distance between each pair of genes across the different time points was computed and a hierarchical clustering algorithm with ward.D linkage was run. The resulting clustering tree was cut at a height of 35, and 9 clusters containing between 383 (cluster 8 and 9) and 669 (cluster 1) genes were retained. (B, C, D) Enriched GO terms in clusters 2, 7, and 9 are depicted as an enrichment map. Using R and the package clusterProfiler terms were filtered by *P*-value < 0.01. Edges between nodes represent overlap between the connected terms and the number of enriched genes found in the respective term are coded by dot size.

as enrichment maps (Fig. 3B through D). Cluster 2 consists of profiles of mutants that display a slight decrease in their representation in the pool during infection, while cluster 7 groups mutants with a more drastic decrease, particularly at 2 × 48 hpi. By contrast, cluster 9 contains mutants that continuously increase in abundance in the pools (Fig. 3A). Clusters 2 and 7 are primarily enriched in genes associated with metabolic processes. Cluster 2 is enriched in pathways related to lipid biosynthesis, including triacylglycerol (TAG) and glycerolipid biosynthetic pathways, and diacylglycerol O-acyltransferase activity (Fig. 3B; Fig. S3A). TAGs have been linked to the persistence of *M. tuberculosis*, serving as a major energy source for slow-metabolizing populations

(56). Within cluster 2, we identified MMAR_1519, the homolog of *tgs1* in *M. tuberculosis*, mutations in *tgs1* in *M. tuberculosis* significantly reduce TAG accumulation (57). Cluster 7 encompasses genes associated with various metabolic processes, including peptidoglycan and PDIM synthesis, fatty acid import (*mce1* operon), amino acid metabolism (glutamine), and vitamin biosynthesis (biotin, cobalamin, and folate) (Fig. 3C; Fig. S3B). In addition, genes involved in DNA repair are prominent in this cluster, indicating that the bacterium needs to repair DNA damage during infection in *D. discoideum*. Counterintuitively, cluster 9 also contained a subset of genes involved in damage DNA binding, possibly indicating that the response to DNA damage is complex and some mutations increase fitness during infection (Fig. 3D; Fig. S3C).

## Fitness cost of mutants during *D. discoideum* infection

To identify mutations that play a role during infection, we compared the mutant abundance (measured by the number of unique Tn-Seq reads per gene) at different infection time points (24 hpi, 48 hpi) and after the second iteration of selection (2 × 48 hpi) with the Inoculum pool. We used the HMM resampling option in the TRANSIT software and considered mutants to be significantly enriched or depleted in a condition if the $\log_2$ ratio was ≥0.585 or ≤−0.585, with a *P*-value of ≤0.05 (Fig. S4E through G; Table S3).

Mutations with no impact on fitness in a given condition have a $\log_2$fc close to zero. A negative $\log_2$fc represents gene disruptions resulting in a fitness disadvantage (FD), while a positive $\log_2$fc indicates gene disruptions leading to a fitness advantage (FA). The results are visualized in volcano plots, displaying the binary logarithmic fold change ($\log_2$fc) and a negative decadic logarithm of *P*-values for each pairwise comparison with the Inoculum (Fig. S4). Overall, we found uneven distributions, with more mutations causing an FD at 24 hpi (Fig. S4E) and 2 × 48 hpi (Fig. S4G), while at 48 hpi (Fig. S4F), the effect of mutations was more evenly spread. Specifically, at 24 hpi, mutations in 70 genes had a significant impact, all leading to an FD. At 48 hpi, 74 mutations had a significant impact, with 45 leading to an FD and 29 to an FA. Finally, at 2 × 48 hpi, the highest number of mutations had a significant impact, with 360 conferring an FD and only 15 leading to an FA.

To examine functional relationships among the enriched and depleted *M. marinum* mutants at different time points, we used the STRING database, which consolidates data from various sources, including physical and functional associations between proteins (58). *M. marinum* genes were mapped to their *M. tuberculosis* orthologs prior to conducting the analysis. A color code was used to indicate the $\log_2$fc, ranging from dark blue (mutations causing FD) to dark red (mutations causing FA). The different conditions were represented as circles around the gene names, with the inner circle corresponding to 24 hpi, the middle layer to 48 hpi, and the outer layer to 2 × 48 hpi (Fig. 4). The analysis confirmed that most mutations predominantly lead to a FD, including mutants in ESX-1-mediated secretion, the OppABCD periplasmic peptide transport system, in iron homeostasis, DNA repair, cobalamin, and biotin metabolism. A few scattered mutations led to an FA, such as in *relA* or Rv3816c.

ESX-1 subunits were identified, forming a high-confidence interactome, including MycP1 (a protease involved in regulating ESX-1 secretion and virulence) and EccCb1 (a member of the FtsK/SpoIIIE-like ATPase family responsible for transporting proteins across the mycobacterial membrane). As visualized by the color-coded rings, most mutations show a monotonous temporal profile. However, some mutations exhibit an alternating phenotype, as seen in *glpK* (FD at 24 hpi and 48 hpi, FA at 2 × 48 hpi), which has been linked to phase variation associated with changes in physiology and drug tolerance during *M. tuberculosis* infection (59). The opposite phenotype was observed for *uvrC* (FA at 24 hpi and 48 hpi, FD at 2 × 48 hpi) a gene involved in the nucleotide excision repair pathway (NER) (Fig. 4).

A GO enrichment analysis was performed to uncover specific pathway signatures and determine a temporal trend in the fitness cost of disrupted genes. We submitted

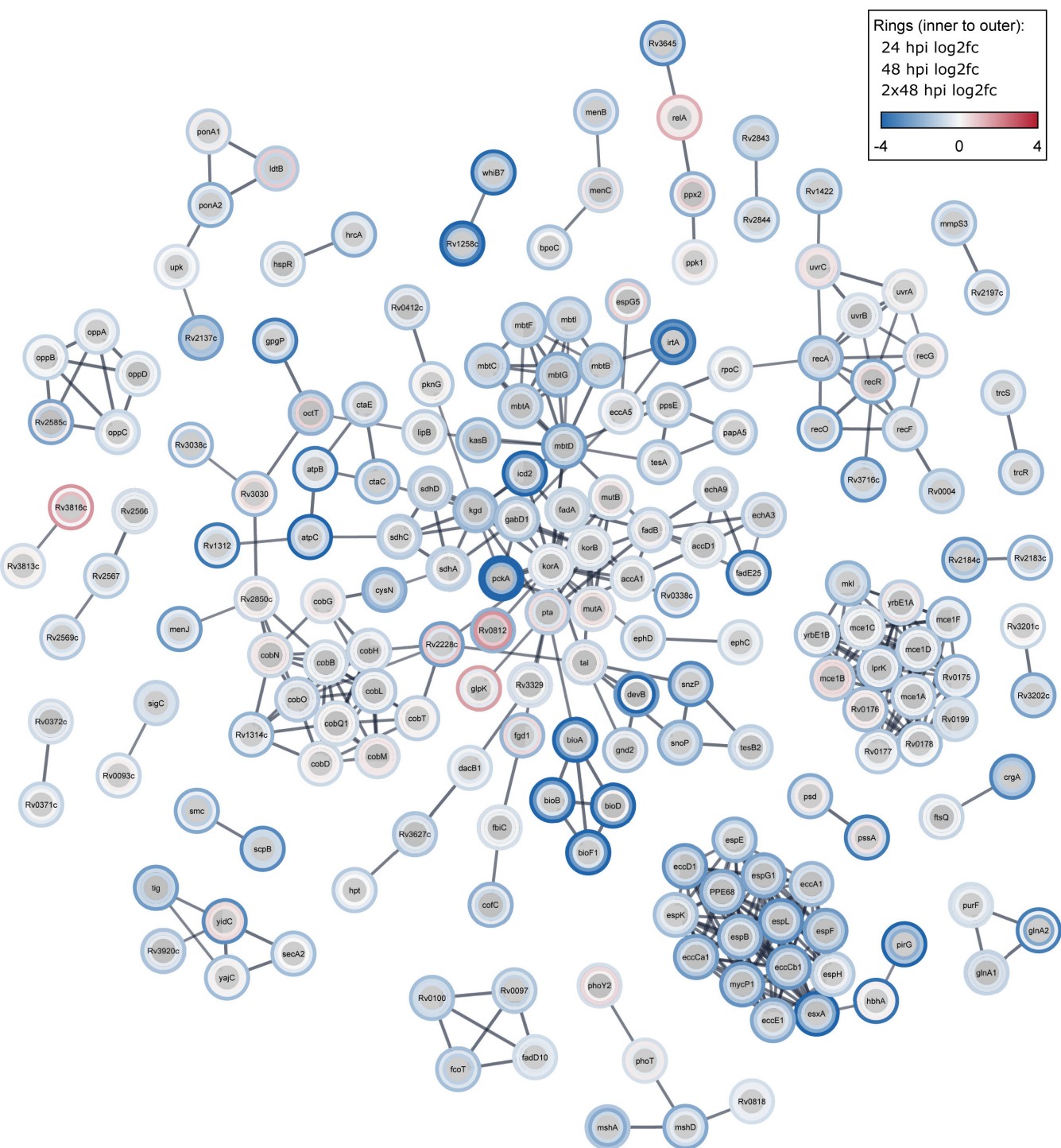

**FIG 4** Protein interactions of FA/FD genes across conditions. The figure displays a protein-protein interaction network of Mtb orthologs of FA or FD genes, retrieved from the STRING database (genes filtered for absolute $\log_2$fc ≥ 0.585, *P*-value ≤ 0.05, projected on the full STRING network, confidence cutoff of 0.8, singletons were omitted, query organism: *Mycobacterium tuberculosis* H37Rv). Color coding was used to identify the effect of the mutation ranging from dark blue (mutations leading to FD) to dark red (mutations leading to FA). The color scale in the layered rings represents from inside to outside the $\log_2$fc of 24, 48, and 2 × 48 hpi, respectively, compared to the Inoculum.

significant genes (*P*-value ≤ 0.1, absolute $\log_2$fc ≥ 0.585) to the enrichment analysis, combining 24 hpi FA and FD genes (Fig. S4A) and 48 hpi FA and FD genes (Fig. S4B; Table S4). We also submitted FA and FD genes separately to the enrichment analysis,

maintaining a *P*-value of ≤0.1 for FA genes (Fig. S4C) and setting a cutoff of *P*-value ≤ 0.01 for FD genes (Table 1; Table S4). Different thresholding was motivated by a more comprehensive visualization, applying a laxer cutoff for categories for early time points and a stricter cutoff for 2 × 48 hpi. The enriched pathways were mainly associated with metabolic processes, cell wall biogenesis, cell cycle, and protein export. The 24 hpi timepoint showed the most limited number of enriched pathways and included the menaquinone pathway (part of the electron transport chain) and the cell division pathway, with genes involved in chromosome structure and partitioning. At the end of one or two rounds of selection (48 and 2 × 48 hpi), coinciding with *M. marinum* cytosolic proliferation, a higher number of functional groups emerged. Among the most enriched pathways common to the 48 and 2 × 48 hpi, we found the pyridoxine (vitamin B6) and the biotin biosynthesis pathways, which support the biosynthesis of methionine, glycine, serine, pantothenate, purines, and thymidine (60). At 2 × 48 hpi, which included the highest number of differentially transposon-mutated genes in the FD category, a cluster of genes related to the SOS response and DNA damage was identified. Many helicases (RecG, MMAR_1358) and the well-studied *recA*, a gene involved in the regulation of nucleotide excision repair and the induction of the SOS response were present, confirming the DNA damage response detected in cluster 7 (Fig. 3C) (61). This reiterates that *D. discoideum* infection results in significant DNA damage in the bacterium and consequently the repair is necessary for intracellular fitness. Interestingly, the fatty acid biosynthesis pathway and related pathways, as shown in the enrichment map, were identified at 2 × 48 hpi in both FA and FD categories (Fig. S4C and D). This indicates the vital role of fatty acid metabolism in mycobacterial survival in the host, consistent with the waxy mycomembrane importance in virulence. Among the genes in these functional groups, we found the polyketide synthase pks15/1 involved in phenolic glycolipid (PGL) synthesis. In addition, the presence of processes linked to fatty acid biosynthesis enriched in both FA and FD categories indicates that the requirements of these metabolic pathways are finely tuned during infection as opposed to growth *in vitro*.

## The *esx-1* operon and genes implicated in vitamin and lipid metabolism are required for *M. marinum* survival in *D. discoideum*

The general pattern highlights that the majority of gene insertions had no impact on intracellular fitness and were considered neutral (NE), with only a few leading to fitness advantages (FA) or disadvantages (FD) in intracellular growth (Fig. 5A through C; Table S3). To visualize the chromosomal distribution of fitness-altering insertions during *D. discoideum* infection and pinpoint potential "hot spots" of significant mutations, we plotted the $log_2fc$ of all genes along their genomic positions. Genes were color-coded based on their *P*-value (*P*-value ≤ 0.1 or *P*-value ≤ 0.01) at 24 hpi (Fig. 5A), 48 hpi (Fig. 5B), and 2 × 48 hpi (Fig. 5C; Table S3).

At 24 hpi, most of the mutations were found in genes of unknown function annotated as conserved or hypothetical proteins. The gene *pckA* (MMAR_0451) encoding phosphoenolpyruvate carboxykinase (PEPCK) and catalyzing the first committed step in gluconeogenesis and the *esx-1* operon (10 genes between MMAR_5440 to MMAR 5449) involved in MCV escape stand out with a relatively high, negative $log_2fc$ and high significance (*P*-value 0–0.01). At 48 hpi, in addition to the continued depletion of mutants in *pckA* and the *esx-1* operon, we observed an increased FD associated with mutations in genes typically required by mycobacteria that are consuming lipids. These included isocitrate lyase *icl2* (MMAR_0158) which is involved in the glyoxylate shunt and the methylcitrate cycle (62), *pckA, korAB* that functions as an alternative α-ketoglutarate dehydrogenase pathway (63), and succinate dehydrogenase (sdhACD) that is required for the further oxidation of the succinate produced by korAB, the glyoxylate shunt, the methylcitrate cycle, and the methyl malonyl pathway. Interestingly, *prpCD,* encoding methylcitrate synthase and methylcitrate dehydratase, are not required for *D. discoideum* infection but the bacterium does need *mutAB* encoded methylmalonyl-CoA

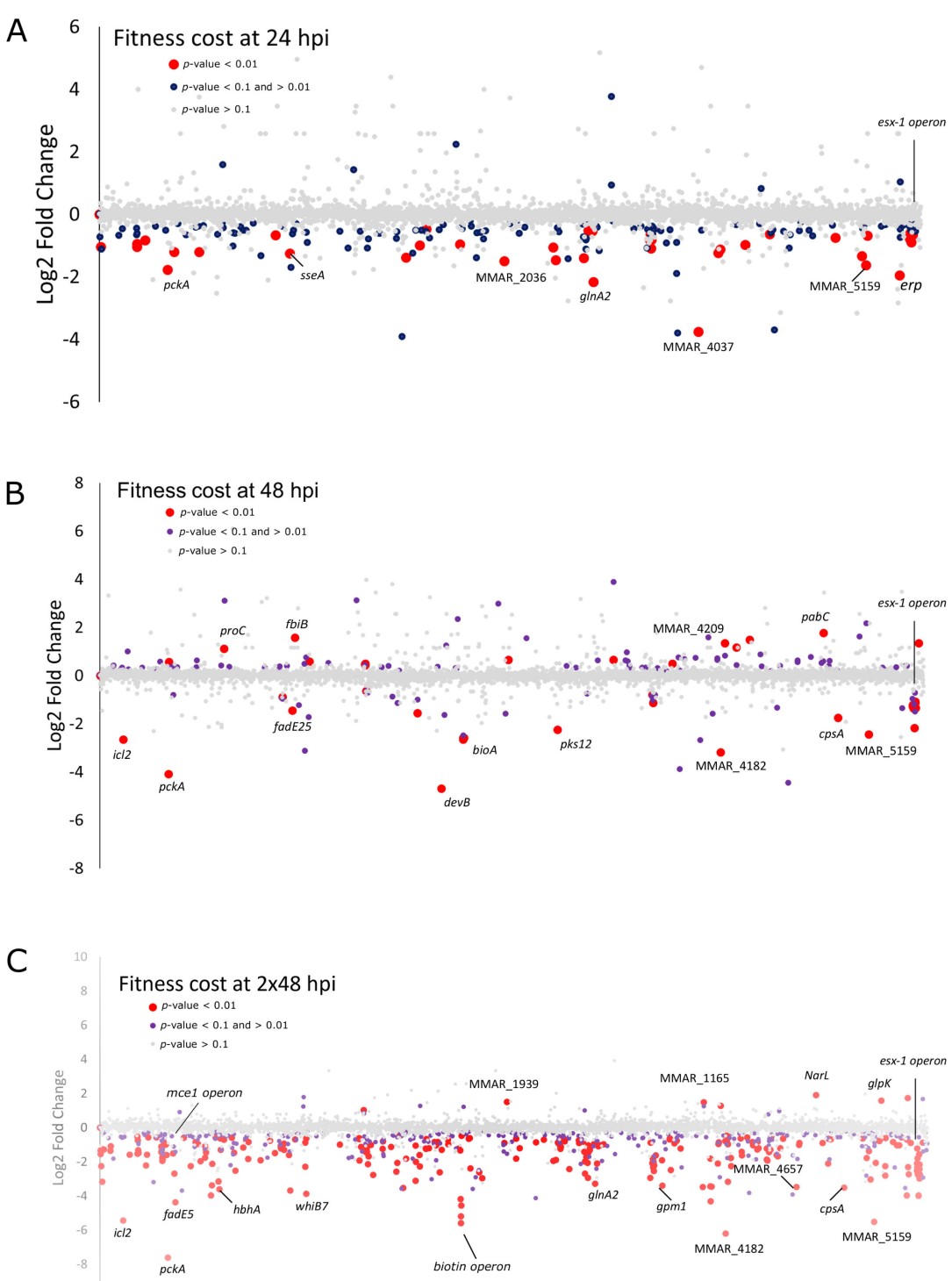

**FIG 5** Fitness cost of *M. marinum* genes in *D. discoideum* over time and with respect to genome position. The log$_2$fc was obtained by comparing the time points 24 hpi, 48 hpi, and 2 × 48 hpi to the Inoculum. Genes are represented by dots which are color-coded according to the *P*-value associated with the log$_2$fc. Red: <0.01, purple: between 0.01 and 0.1, gray: >0.1. Plotting the log$_2$fc in dependence of genomic position (*x*-axis) reveals the fitness impact of operons in each condition: (A) 24 hpi, (B) 48 hpi, and (C) 2 × 48 hpi. Striking loci and operons are highlighted by labels and circles, such as the *esx-1* and *mce1* operons or genes associated with biotin metabolism.

mutase implying bypass of the propionate half of the methylcitrate cycle using the methylmalonyl pathways. This activity requires biotin and cobalamin; thus, this is one

mechanistic reason for the essentiality of biotin and cobalamin synthesis for intracellular infection (Fig. 4; Table S3). A further prominent attenuating mutation was observed in *cpsA* (MMAR_4966), which encodes a LytR-CpsA-Psr (LCP) domain-containing protein necessary for *M. tuberculosis* to evade killing by NADPH oxidase and LC3-associated phagocytosis (64). Interestingly, some mutations at this time point conferred a fitness advantage. For instance, insertions in the genes encoding the ABC transporter *pabC* (MMAR_4873) and *proC* (MMAR_0826), as well as *fbiB* (MMAR_1280), functions of which are not fully elucidated.

At 2 × 48 hpi, mutations in *icl2*, *pckA,* and the *esx-1* operon further reduced the fitness compared with 48 hpi. The number of genes in the *esx-1* operon with mutations exhibiting significant FD increased from 10 genes at 48 hpi to 16 at 2 × 48 hpi. Some other well-known virulence factors were implicated at various stages of infection (Table S2), including the heparin-binding hemagglutinin (MMAR_0800: *hbha*), responsible for bacterial adhesion to non-phagocytic host cells in vertebrate infection, and a subunit of the T7SS ESX-5 (MMAR_2680), responsible for transporting cell envelope proteins. Mutants in various genes implicated in lipid metabolism (MMAR_1778: *tesA*, MMAR_2124: *relA*, MMAR_4966: *cpsA*, MMAR_0258: *fadD10*, MMAR_0319: *fadD7*, MMAR_4864: *mshD*), in the *whiB7* transcriptional regulatory factor (MMAR_1365), and *pknG,* encoding a serine/threonine protein kinase crucial for *M. tuberculosis* viability *in vitro* and infection models (65) were also depleted.

Furthermore, mutations in genes involved in vitamin metabolism, such as biotin (MMAR_2383 to MMAR_2387) or cobalamin (MMAR_1883 to MMAR_1885), also led to a strong FD. In terms of mutations resulting in an FA, we observed a depletion of mutants in gene encoding a nitrate response regulator (MMAR_4782: *NarL*), a glycerol kinase, as discussed above (MMAR_5208: *glpK*), an acetyltransferase (MMAR_5379), and several hypothetical secreted or regulatory proteins (MMAR_1939, MMAR_1165) (Table S3). The *glpK* phenotype likely results from the change of carbon source between the Middlebrook media and the intracellular niche.

## Fitness cost of mutations in the *mce1* and *mce4* operons during *D. discoideum* infection

In bacteria, functionally related genes are often organized into operons, ensuring the coordinated expression of all constituent genes and the proper stoichiometry of the corresponding proteins and enzymes. As mentioned earlier, we observed pathways linked to lipid biosynthetic processes in both FA- and FD-enriched GO terms (Fig. S4C and D). This raised the question of whether other operons, as organizational units, exhibit similar trends in terms of fitness cost (FA or FD). We investigated the impact of mutations in key mycobacterial operons, including the canonical virulence locus *esx-1* (Fig. 6A), the two lipid transporter operons *mce1* and *mce4* (Fig. 6B), and genes associated with PDIM and PGL biosynthesis (Fig. 6C) (7, 66). The analysis of the *esx-1* operon and its associated genes revealed that most mutations led to FD, underscoring the indispensable role of this secretion system for *M. marinum* growth during infection. It is noteworthy that the fitness deficit conferred by most of the *esx-1* mutations increased during the course of infection and upon iteration of the selection, with a 16-fold difference observed for mutation in genes encoding EsxA (ESAT-6) and EsxB (CFP10) at 2 × 48 hpi (Fig. 6A; Table S3).

During infection, mycobacterial survival heavily relies on importing lipids from the host via Mce transporters to support central metabolism and cell-wall biosynthesis (60). Heatmaps showcasing the fitness trends of mutations in the *mce1* operon, responsible for fatty-acid transport, and the *mce4* operon, responsible for (chole)sterol transport, are shown in Fig. 6B. Both Mce systems consist each of two putative permeases (*mce1*: MMAR_0410, MMAR_0411 – *mce4*: MMAR_4989, MMAR_4988) and six cell-wall-associated proteins (*mce1*: MMAR_0412 to MMAR_0417 - *mce4*: MMAR_4982 to MMAR_4987). Genes encoding for accessory proteins such as *LucA* (MMAR_5242/Rv3723), Mce-associated membrane proteins (MAMs), an orphaned MAM (OMAM), and *mceG* (MMAR_0439/

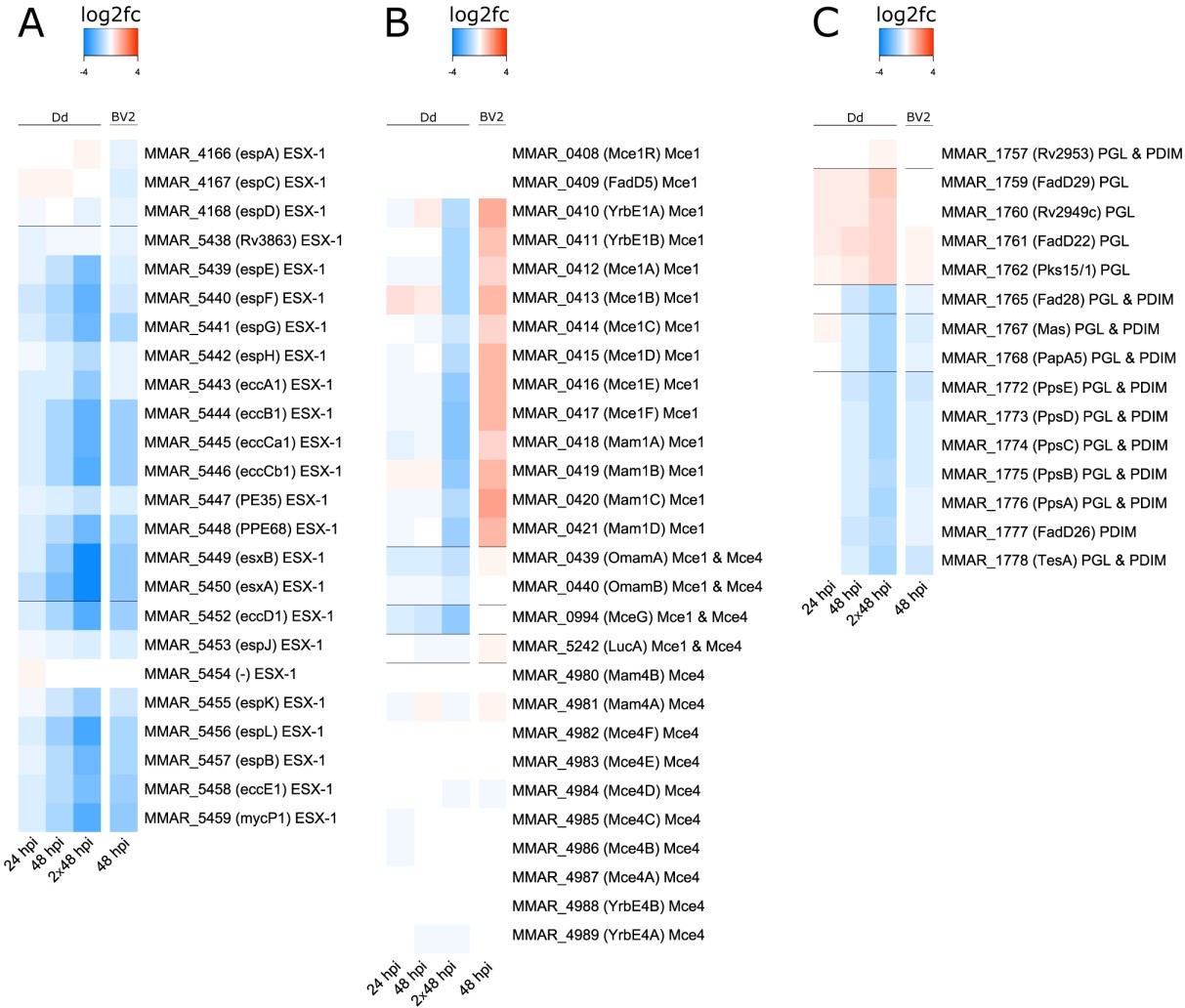

**FIG 6** Fitness cost over time for selected genes and operons of *M. marinum* in *D. discoideum*. The color-coded log$_2$fc in dependence on conditions reveals fitness costs within operons. Operons and associated genes were selected. (A) *esx-1* operon system, (B) *mce1* and *mce4* operon systems, (C) genes associated with phthiocerol dimycocerosates (PDIM) and phenolic glycolipids (PGL) synthesis. The labels include the *M. marinum* MMAR nomenclature, in brackets the Mtb ortholog (as indicated on mycobrowser.epfl.ch, with "-" indicating the absence of an Mtb ortholog), and the associated function.

Rv0199), encoding a putative ATPase, play crucial roles in stabilizing and facilitating the function of both Mce transporters (Fig. 6B) (19, 67, 68) (Fig. 6B). Note that only the *mce1* operon has a regulator encoded by *mce1R* (MMAR_0408).

Our data revealed that all mutations in genes belonging to the *mce1* operon led to FD, beginning at 48 hpi and reaching their highest negative fold change at 2 × 48 hpi. The only exceptions were Mce1R (MMAR_0408) and Fad5 (MMAR_0409), which exhibited a neutral phenotype. Most importantly, mutations in the gene encoding the ATPase MceG, predicted to interact with all the Mce transporters and to be the only cytoplasmic protein, led to an FD (69). These results suggest that mutations in the *mce1* operon and accessory genes primarily affect *M. marinum* growth and survival during the late cytosolic stage of infection in *D. discoideum* (Fig. 6B). However, mutations in the *mce4* operon displayed irregular phenotypes, with some mutations causing a neutral effect or slight FD. This finding is intriguing, as mutations in the *mce4* operon, previously described as detrimental to *M. tuberculosis* survival in the host, appear to have a relatively minor impact on the bacterium during infection in *D. discoideum*. Interestingly, mutations in two genes encoding OMAMs, *omamA* and *omamB*, resulted in a progressive

FD between 24 and 2 × 48 hpi (Fig. 6B), likely because of their function in Mce1-mediated transport.

The pathway that leads to PDIM and PGL synthesis comprises common steps leading to mycocerosic acid (PGL&PDIM, Fig. 6C), followed by two specific branches (PDIM or PGL). PDIMs play a pivotal role in bacterial entry into the macrophage, the regulation of phagosome acidification, phagosomal escape, and the utilization of host cholesterol as a carbon source (6, 70). On the other hand, PGLs are polyketide-derived virulence factors capable of limiting the capacity of activated macrophages to induce nitric oxide synthase (iNOS) and generate NO upon mycobacterial infection (71). In contrast to the *mce1* and *mce4* operons, almost all mutations in genes common to the biosynthesis of PGL and PDIM biosynthesis exhibited a fitness defect at 48 hpi and 2 × 48 hpi (Fig. 6C), supporting the notion that both may play a pivotal role at late stages of infection (10, 11). A closer examination of PDIM and PGL pathways indicated a fitness defect for PDIM mutants, while mutations in genes specific to PGL biosynthesis (including the operon MMAR_1759 to MMAR_1762) led to an intracellular fitness advantage phenotype. This counterintuitive positive impact of loss of PGL synthesis might be resulting from an indirect gain of function due to substrate rerouting to the PDIM pathway upon interruption of the PGL pathway, increasing total PDIM synthesis, boosting virulence and thus effectively leading to an apparent fitness advantage of these transposon mutants.

## Comparative Tn-Seq shows common fitness impact and essentiality in amoebae and murine microglial host cells during *M. marinum* infection

To provide a comprehensive comparison and highlight evolutionary conservation, we conducted Tn-Seq experiments with *M. marinum* during infection of BV2 murine microglial cells. A single time point of 48 hpi was selected because, as in *D. discoideum,* a complete infection cycle in these phagocytes follows the same stages and also lasts approximately 48 hours (see Fig. 1A) (13, 29, 72). In total, seven samples were included in the analysis: four from the inoculum and three at 48 hpi (Table S2). Subsequently, a PCA was computed and plotted to observe the data distribution, revealing distinct clustering of the inoculum and 48 hpi samples (Fig. S5).

To identify mutations impacting intracellular bacterial growth during BV2 infection, we applied the TRANSIT resampling method to compare read counts at 48 hpi to those of the inoculum, analogous to the procedure used in the *D. discoideum* experiments. An examination of all genes on a volcano plot (Fig. 7A) clearly illustrated that there were more mutations leading to an FD compared to those leading to an FA. This trend was consistent with the results obtained for the 2 × 48 hpi *D. discoideum* infection (Fig. S4G). Notably, the mutation with the highest FD (a 32-fold change) was found in a gene coding for glucose-6-phosphate isomerase Pgi (MMAR_4557), while the mutation with the highest FA (a 10-fold increase) was found in the aspartate decarboxylase PanD (MMAR_5104). PanD is essential for the biosynthesis of coenzyme A and is the putative target of the first-line *M. tuberculosis* antibiotic, pyrazinamide. The apparent intracellular growth advantage of *panD* mutation in BV2 microglial cells may indicate the increased availability of pantothenate or β-alanine in these cells relative to the microbiological medium.

Next, a GO enrichment analysis was performed on differentially represented mutations at 48 hpi, using *P*-value ≤ 0.1 and absolute $\log_2$fc ≥ 0.585. Enriched terms were then filtered by *P*-value < 0.01 (Fig. 7B and C). Similar functional groups were enriched during BV2 infection compared to *D. discoideum*, with a prominent focus on vitamin biosynthetic processes such as for biotin and cobalamin (Table S4). The enrichment map also revealed an rRNA methylation cluster, hinting at ribosomal heterogeneity and altered translation during infection (Fig. 7B and C). Certain functional groups enriched after the selection in amoebae, such as genes from the TCA cycle and lipid catabolic processes, were not evident in BV2 cells indicating metabolic differences of bacteria internalized in amoeba and microglial cells (Table S4).

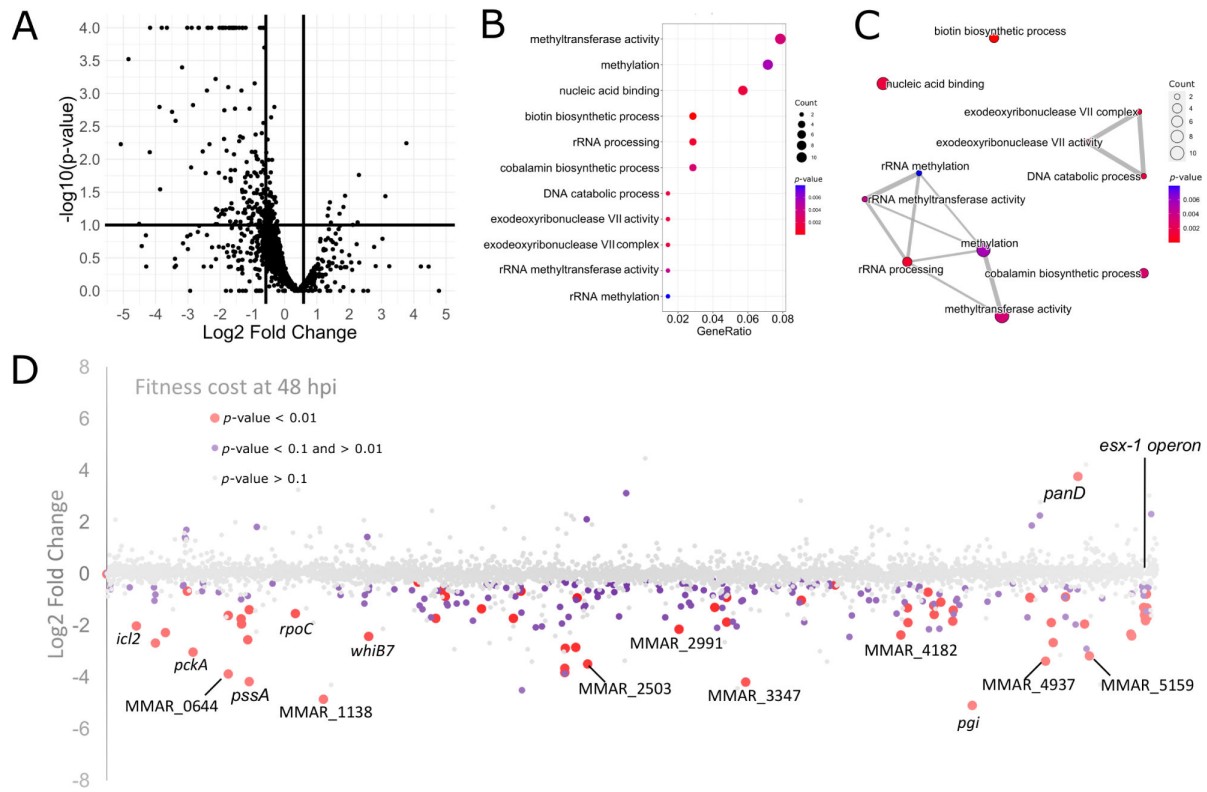

**FIG 7** Fitness cost over time of *M. marinum* in BV2 microglial cells. (A) Volcano plot of 48 hpi versus the inoculum. The *x*-axis shows the $\log_2$fc and the *y*-axis the $-\log_{10}$ of the associated *P*-values. Thresholds used to filter genes for enrichment analysis are depicted as thick black lines: $\log_2$fc ≥ 0.585 and ≤0.585 (corresponding to 1.5-fold change) and *P*-value ≤ 0.1. For visualization, *P*-values < 0.0001 were capped at 0.0001. (B, C) Thresholded genes at 48 hpi were submitted to enrichment analysis using R and the package clusterProfiler. Terms were filtered by *P*-value < 0.01. In (C), edges between nodes represent the overlap between the connected terms, enriched genes found in the respective terms are coded by dot size. GeneRatio equals the number of differentially expressed genes against the number of genes associated with a GO term in the *M. marinum* genome. (D) Fitness cost at 48 hpi in BV2 microglial cells and with respect to genome position. Genes are represented by dots which are color-coded according to the *P*-value associated with the $\log_2$fc. Red: <0.01, purple: between 0.01 and 0.1, gray: >0.1. Plotting the $\log_2$fc in dependence on genomic position (*x*-axis) reveals the fitness impact of operons at 48 hpi.

As in the *D. discoideum* infection, the $\log_2$fc values were plotted along the chromosome to identify spatial relationships of intracellular fitness genes, with significance coded by *P*-value ≤ 0.1 or *P*-value ≤ 0.01 (Fig. 7D). These plots revealed no major patterns associated with genomic position although they highlighted the importance of the region encoding ESX-1. Among the mutations leading to FD, we identified genes like *icl2*, *pckA*, and *whiB7*, associated with lipid metabolism as well as cell wall biosynthesis (MMAR_1778: *tesA*), and survival during infection (MMAR_0713: *pknG*), and the *esx-1* operon, all of which play a crucial role in the bacterium's survival in both hosts (see "BV2" column in Fig. 6A). However, some mutations leading to an FD were specifically identified during BV2 infection, such as MMAR_0644, MMAR_1138, MMAR_2991, MMAR_3347, and MMAR_4937. Their functions remain undescribed, except for glucose-6-phosphate isomerase (MMAR_4557: *pgi*), which plays a central role in glycolysis and gluconeogenesis and its disruption results in glucose auxotrophy (73), and trehalose-6-phosphate synthetase (MMAR_4978: *otsA*), which is required for optimal growth *in vitro* and survival in a mouse *M. tuberculosis* infection model (74). As in *D. discoideum* infections, mutations in the *mce4* operon and genes encoding accessory proteins do not lead to FD or FA (Fig. 6B), while most mutations in the *mce1* operon led to opposite phenotypes, an FD in the amoeba and an FA in BV2 (though not reaching statistical significance) (Table S3).

## Conclusions

The combination of transposon insertion mutagenesis and deep sequencing allows for quantifying insertion mutants in a pool, aiding in the identification of essential genes and those linked to selectable phenotypes (37). In this study, we employed Tn-Seq to assess mutation fitness costs in *M. marinum*, aiming to understand host-pathogen interactions in *D. discoideum* and BV2 microglial cells. Our findings revealed that 57% of TA sites in the *M. marinum* genome were disrupted, with 9.2% at low-permissive insertion sites, consistent with previous Tn-Seq studies in pathogenic mycobacteria (44, 75, 76).

Specifically, our results were similar to Tn-Seq studies in *M. tuberculosis*, where 42%–64% of TA sites and 9% of low-permissive sites were disrupted (49). Our study identified 568 essential genes (10.2%), comparable to previous Tn-Seq studies in *M. tuberculosis* (40, 77). Unlike an earlier Tn-Seq study on *M. marinum* E11, which identified only 6% of essential genes (45), the discrepancy might be due to genomic diversity as *M. marinum* strains were classified into two different clusters "M" and "Aronson" types (78). Among the essential genes identified in *M. marinum* M, 237 orthologs were found in *M. marinum* E11 (37.3%) (45), 311 orthologs in *M. tuberculosis* (44.4%) (40), and 161 orthologs in *M. bovis* (42). The larger overlap of *M. marinum* with *M. tuberculosis* H37Rv compared to the E11 strain of the same species might be partly due to *M. marinum* M strain and *M. tuberculosis* H37Rv being able to infect humans, whereas *M. marinum* strain E11 has more environmental hosts. Our study identified 153 essential genes specific to the *M. marinum* M strain. This specificity could result from variations in sequencing depth or species-specific genomic differences. Our study, along with others, demonstrates that Tn-Seq is a robust technique for identifying pathways affecting bacterial fitness during infection. However, pooled strain studies are susceptible to trans-complementation or "helper effects," particularly in clumpy mycobacteria, where co-infection of host cells can potentially lead to false-positive or false-negative fitness impact results.

The importance of vitamin metabolism for *M. marinum* during *D. discoideum* infection was highly evident in our experiments. Biotin, in particular, emerged as essential for *M. marinum* survival in both cell types, leading to a pronounced fitness deficit when mutations occurred in genes such as MutA/B and BioA/B/F1/D. Biotin is an indispensable cofactor for biotin-dependent enzymes involved in the catabolism of fatty acids via MutA/B in the TCA cycle (79). Interestingly, *M. tuberculosis* cannot scavenge sufficient biotin from its host and relies on *de novo* synthesis through genes like *bioA* and *bioD*. This suggests that biotin may be universally required for *M. marinum* during eukaryotic cell infection, in line with previous studies indicating the role of biotin in establishing and maintaining chronic infections in murine tuberculosis models (80). Carbon metabolism, including glycolysis and gluconeogenesis, also emerged as essential pathways for *M. marinum* survival. Mutations in genes related to these pathways, such as *pckA* (encoding the first enzyme in gluconeogenesis), were detrimental during infection and essential for establishing and maintaining the infection (81). Furthermore, the data underscored the significance of the methylcitrate cycle in mycobacteria infection, as mutations in *icl* also resulted in a fitness deficit during *D. discoideum* infection (53). This is again indicative that the intracellular bacterium is utilizing lipids as a nutritional source. Altogether, these findings validate the robustness of the Tn-Seq method in *M. marinum* for highlighting important pathways involved in bacterial survival during infection. Furthermore, these results demonstrate the conservation of virulence strategies employed by *M. marinum* to infect amoebae and mammalian cells.

However, some differences were observed between the *D. discoideum* and BV2 models. For instance, genes like HBHA, linked to adherence and lipid inclusion formation, were required only for *D. discoideum* infection (82, 83). Similarly, mutations in genes encoding enzymes involved in mycobactin production, a siderophore produced by the bacteria to scavenge iron, only led to an FD during *D. discoideum* infection, although they are known to affect bacterial virulence in human hosts, too (84, 85). Differences between the two host phagocytes included the observation that mutation of the RelA GTP pyrophosphokinase, which controls the stringent response, was only found to be

attenuating in BV2 cells. This may reflect a fundamental difference in growth control required for survival in the different host cells. These potential host specificities might be attributed to differences in selection timing, the temperature during infections, and/or distinct host defense or pathogen virulence strategies.

Our Tn-Seq data demonstrate that mutations occurring in many sites of the *mce1* operon in the *D. discoideum* model resulted in an FD, whereas mutations in the same operon had a neutral effect during BV2 microglial cell infection (Fig. 6B). Moreover, mutations in genes encoding proteins involved in the stability of both the Mce1 and Mce4 complexes, such as the accessory OmamA protein (68) and the ATP-ase MceG, lead to an FD in *D. discoideum* only, while mutations in LucA had neutral effect in both phagocytic models (67). These results support the hypothesis outlined in the literature that mutations in the *mce1* operon impact fatty acid acquisition and result in reduced intracellular growth of *M. tuberculosis*, although a direct link between lipid acquisition and intracellular growth is not completely understood. One possible explanation for the variation in fitness phenotypes between these phagocytes might lie in the differences in lipid composition between *D. discoideum* and BV2 microglial cells. *M. marinum*'s ability to infect a wide range of hosts allows it to adapt its lipid metabolism based on the lipids available in the infected environment. To investigate this hypothesis, further investigations are required to understand the impact of *mce1* mutations during the course of infection.

Our data (Table S3) align closely with the results of numerous targeted and genome-wide mutagenesis studies in *M. marinum*, particularly those investigating fitness costs during infection in both amoebal and animal phagocytes. Our Tn-Seq data clearly indicate an FD as a result of mutations in the *esx-1* locus, consistent with reports that the RD1 and DCE mutants are attenuated for intracellular growth in both *D. discoideum* and macrophages (86, 87). Another study examining the role of TesA in cell-wall-associated lipid synthesis and intracellular growth found that a *tesA* mutant is attenuated in both *D. discoideum* and zebrafish embryos (88), in line with the present genome-wide analysis.

At $2 \times 48$ hpi, mutations in genes encoding some putative effectors of the *esx-5* operon, including *esxM*, led to an FD, which correlates with previous publications showing that an *esxM* knockout is attenuated during infection (89). Similarly, mutations in iron acquisition genes, *mbtB* and *irtB,* resulted in a fitness defect, which corroborates reports for $\Delta mbtB$ and $\Delta mbtB$-$\Delta irtAB$ strains being impaired for growth in various phagocytes, including *D. discoideum* (84). It is worth noting, however, that some mutations did not impact infection, including in the zinc efflux gene *ctpC*, while others in genes encoding PIP phosphatases (PtpA, PtpB, and SapM) led to an FA, whereas recent publications have shown that CtpC (27) and PIP phosphatases (90) are required for intracellular growth of *M. marinum* in *D. discoideum*.

In conclusion, this study presents a successful Tn-Seq analysis performed in *M. marinum* M, providing a robust resource to explore the fitness costs of gene disruption during infection. We have identified genes and pathways that contribute to *M. marinum* infectivity, which can be further investigated in future studies. Collectively, this study opens the door for the use of Tn-Seq in rapidly screening and identifying genes involved in *M. marinum* fitness across various host model systems, offering diverse insights into the evolution of pathogenicity in mycobacteria.

## MATERIALS AND METHODS

### Bacterial strains, plasmids, and culture methods

Bacterial strains used in this study are listed in Table 1. *M. marinum* M strain was grown in Middlebrook 7H9 (Difco) supplemented with 10% OADC (Becton Dickinson), 0.2% glycerol (Biosciences), and 0.05% Tween 80 (Sigma) at 32°C in shaking culture at 150 rpm in the presence of 5 mm glass beads to prevent aggregation. The axenic *D. discoideum* Ax2(Ka) was cultured at 22°C in Hl5c medium (Formedium) supplemented with 100 U/mL of penicillin and 100 µg/mL of streptomycin (Invitrogen). BV2 cells were grown at 37°C

**TABLE 1** Thresholding of the input for pathway analyses[a]

| Host organism | Condition | FA & FD | FA | FD | Thresholds |
|---|---|---|---|---|---|
| *D. discoideum* | 24 hpi | 97 | 7 | 90 | abs(log$_2$fc) ≥ 0.585; *P*-value ≤ 0.1 |
| | 48 hpi | 97 | 38 | 59 | abs(log$_2$fc) ≥ 0.585; *P*-value ≤ 0.1 |
| | 2 × 48 hpi | 268 | – | – | abs(log$_2$fc) ≥ 0.585; FA & FD: *P*-value ≤ 0.01 |
| | – | – | 19 | – | abs(log$_2$fc) ≥ 0.585; FA: *P*-value ≤ 0.1 |
| | – | – | – | 261 | abs(log$_2$fc) ≥ 0.585; FD: *P*-value ≤ 0.01 |
| BV2 | 48 hpi | 183 | 12 | 171 | abs(log$_2$fc) ≥ 0.585; *P*-value ≤ 0.1 |

[a]–, not applicable.

in Dulbecco's modified Eagle's medium (DMEM) high glucose supplemented with 6% of heated inactivated fetal bovine serum (FBS) and 100 U/mL of penicillin in a 5% $CO_2$ environment.

## Selection of transposon *M. marinum* mutants in *D. discoideum* and BV2

A transposon mutant library was generated essentially as described previously (41, 42). Briefly, *M. marinum* was grown in 300 mL to the mid-log phase at 30°C. The bacteria were centrifuged at 4,000 rpm and washed with pre-warmed MP buffer (50 mM Tris-HCl, pH 7.5, 150 mM NaCl, 10 mM $MgSO_4$, and 2 mM $CaCl_2$). The cells were resuspended in a total volume of 10 mL MP buffer and 5 mL of fMycoMar phage stock at approximately $1 \times 10^{11}$ cfu/mL was added and incubated for 6 hours at 32°C. The transduction mix was centrifuged and washed in 7H9 with 0.05% Tween 80 and plated on 14 × 15 cm petri dishes containing 7H11/OADC with kanamycin at 25 µg/mL and grown at 30°C for 7 days before harvesting into 10% glycerol and freezing at −80°C. Dilutions of the transduction were simultaneously plated for enumeration. For library selection, the *M. marinum*-Tn library inoculum was spinoculated on $5 \times 10^6$ adherent *D. discoideum* cells at a multiplicity of infection (MOI) of 5 to minimize any helper effect by having 1 to 3 bacteria per cell. After washing off non-phagocytosed bacteria, the infected cells were incubated at 25°C in a filtered Hl5c medium containing 5 µg/mL of streptomycin and 5 U/mL of penicillin to avoid extracellular growth. At either 24 or 48 hpi *D. discoideum* were scraped and lysed with Triton 0.1%, intracellular bacteria were collected and plated on ten 15 cm 7H10 plates with 25 µg/mL kanamycin, 10% OADC, 0.2% glycerol, and 0.05% Tween-80. The same incubation time was applied after each collected time point. The same procedure was used for the second round of selection but the inoculum used for this step was the frozen aliquot of *M. marinum* collected at 48 hpi after the first round.

## BV2 cells infection assay

BV2 cells were cultured in DMEM medium supplemented with FCS in 10 cm petri dishes at 37°C. The day prior to infection, cells were trypsinized, and passaged to reach 60–70% confluency on the day of the experiment. For library selection, the *M. marinum*-Tn library inoculum was spinoculated on $5 \times 10^6$ adherent BV2 cells at a multiplicity of infection (MOI) of 3 to minimize any helper effect. Mycobacteria were centrifuged at RT at $500 \times g$ for 10 min onto pre-plated BV2 cells to promote efficient and synchronous uptake. After 20 min of phagocytosis, extracellular bacteria were washed and cells were incubated for 48 hours under 5% of $CO_2$ at 32°C. At 48 hpi, BV2 cells were scraped and lysed with Triton 0.1%, intracellular bacteria were collected and plated on ten 15 cm 7H10 plates with 25 µg/mL kanamycin, 10% OADC, 0.2% glycerol, and 0.05% Tween-80.

## Construction of gDNA libraries of *M. marinum* mutant pools

*M. marinum* gDNA was extracted following an adapted protocol from Belisle and Sonnenberg (91). Briefly, *M. marinum* mutant pools were centrifuged and resuspended in breaking buffer (50 mM Tris, 10 mM EDTA, and 100 mM NaCl, pH8). Mechanical lysis was performed with the Fastprep homogenizer for 30 s, this step was repeated three times. The upper phase was chemically lysed with RNase (200 µg/mL) and lysozyme

(100 µg/mL) 1 hour at 37°C. Then, 0.1 volume of SDS 10% and 0.01 volume of proteinase K (100 µg/mL) were added for 1 h at 55°C. Extraction steps were realized with an equal volume of phenol:chloroform:isopropanol 25:24:1 for 30 min followed by chloroform:iso-propanol for 5 min. Precipitation was performed with 3 M sodium acetate pH 5.2 and 1 volume of isopropanol 100%, gDNA was pelleted and washed with ethanol 70%. The extracted gDNA was quantified using Qubit (Invitrogen) and its quality was checked with the TapeStation system (Agilent).

A 2 µg aliquot of gDNA in Tris buffer (10 mM Tris, 0.1 mM EDTA, pH 8) was sheared in a Covaris S2 machine following these parameters: intensity 4, duty cycle 10%, cycles 200 and time 65 s. After purification with AMPure XP purification kit (Agencourt), 1,000 ng of sheared DNA fragment ends were repaired, blunt ended and a d-A tail added following NEBNextEnd/d A-Tailing module protocol. Specific adapters 1 and 2 described in Table 2 were self-hybridized and then ligated to the "A" tail ends by T4 ligase (Promega). After purification with AMPure XP purification kit (Agencourt), 40 ng of libraries was amplified by PCR using KAPA polymerase with the following parameters: 95°C 3 min; 98°C 20 s; 60°C 15 s, 72°C 15 s (repeated for 25 cycles); 72°C 1 min for elongation. Transposon junctions were amplified with the IS6 primer: 5′ CAAGCAGAAGACGGCATACGA 3′ and specific primers MarA to MarP described in Table 3. PCR products were purified with the AMPure XP purification kit and 16-plexed samples were pooled in approximately equimolar amounts of 5 nM and run in single and double index reads on a Hiseq 4000 (Illumina).

## Transposon sequence analysis

After demultiplexing the sequence data with the P5-index reads, the first step was pre-processing of the obtained raw sequences in the fastq format. The raw sequence files were first checked using the FASTQC tool which allows detection of low-quality samples and outliers via sequence quality plots. For each read, the read end nucleotides with a FASTQ quality Phred score "#" were removed, and the whole read was disposed of if the number of "#" exceeded 50 bp of half read length. Also, non-transposon reads were filtered out by keeping only the reads containing the sequence "TGTTA" at the beginning of the read. After these pre-processing steps, reads were mapped on the reference genome, which consists of the sequence of *M. marinum* M genome (NC_010612.1) and *M. marinum* M plasmid pMM23 (NC_010604.1) using Bowtie2. Only alignments with mapping quality >30 were kept for the downstream analysis which consisted of counting transposon insertions from alignment SAM files. Alignments having both the same insertion site and P7-indexed read were considered artefactual amplicons generated during the PCR amplification and were subsequently removed prior to counting of transposon insertions for each TA site. To detect and flag low-permissive sites, we determined the surrounding ±3 bp sequence and checked whether it matches with the pattern "SGNTANCS," which has been associated with decreased transposon insertion (49) (see Table S1 at https://tinyurl.com/5dbxww5w).

The well-documented open-source software TRANSIT (50) was used for essentiality analysis and conditional essentiality analysis. For the former, the hidden Markov model (HMM) method was used with default parameters. The method uses an HMM to classify both, TA sites and corresponding genes into essential, non-essential, growth defects or growth advantages. NC_010612.1 and CP000854.1 were both used as gene annota-tions, plus the plasmid M plasmid pMM23 NC_010604.1. The TRANSIT input and output are documented in Tables S1 (https://tinyurl.com/5dbxww5w) and S2. TRANSIT outputs summary files on the TA site level (Table S1) and the gene annotation site level (Table S2). On the TA site level, only unique TA sites are presented, whereas in the gene summary

**TABLE 2** Bacterial strains used in the study

| Strains | Comments | Source or reference |
|---|---|---|
| *M. marinum* strain M | WT strain | TMC1218 |
| *M. marinum* strain M | Kan[R] | This study |

**TABLE 3** Primers used in the study for *M. marinum* mutant pools amplification

| Name | Sequences (5′–3′) |
| --- | --- |
| Adapt1 | c*a*a*g*cAGAAGACGGCATACGAGATNNNNNNNGTGACTGGAGTTCAGACGTGTGCTCTTCCg*a*t*c*t |
| Adapt2 | g*a*t*c*gGAAg*a*g*c*a–P |
| IS6 | CAAGCAGAAGACGGCATACGA |
| MarA | AATGATACGGCGACCACCGAGATCTACACATCACGACACTCTTTCCCTACACGACGCTCTTCCGATCTCGGGGGACTTATCAGCCAACC |
| MarB | AATGATACGGCGACCACCGAGATCTACACCGATGTACACTCTTTCCCTACACGACGCTCTTCCGATCTTCGGGGGACTTATCAGCCAACC |
| MarC | AATGATACGGCGACCACCGAGATCTACACTTAGGCACACTCTTTCCCTACACGACGCTCTTCCGATCTGATACGGGGACTTATCAGCCAACC |
| MarD | AATGATACGGCGACCACCGAGATCTACACTGACCAACACTCTTTCCCTACACGACGCTCTTCCGATCTTATCTACGGGGACTTATCAGCCAACC |
| MarE | AATGATACGGCGACCACCGAGATCTACACACAGTGACACTCTTTCCCTACACGACGCTCTTCCGATCTCGGGGACTTATCAGCCAACC |
| MarF | AATGATACGGCGACCACCGAGATCTACACGCCAATACACTCTTTCCCTACACGACGCTCTTCCGATCTTCGGGGACTTATCAGCCAACC |
| MarG | AATGATACGGCGACCACCGAGATCTACACCAGATCACACTCTTTCCCTACACGACGCTCTTCCGATCTGATACGGGGACTTATCAGCCAACC |
| MarH | AATGATACGGCGACCACCGAGATCTACACACTTGAACACTCTTTCCCTACACGACGCTCTTCCGATCTTATCTACGGGGACTTATCAGCCAACC |
| MarI | AATGATACGGCGACCACCGAGATCTACACGATCAGACACTCTTTCCCTACACGACGCTCTTCCGATCTCGGGGACTTATCAGCCAACC |
| MarJ | AATGATACGGCGACCACCGAGATCTACACTAGCTTACACTCTTTCCCTACACGACGCTCTTCCGATCTTCGGGGACTTATCAGCCAACC |
| MarK | AATGATACGGCGACCACCGAGATCTACACGGCTACACACTCTTTCCCTACACGACGCTCTTCCGATCTGATACGGGGACTTATCAGCCAACC |
| MarL | AATGATACGGCGACCACCGAGATCTACACCTTGTAACACTCTTTCCCTACACGACGCTCTTCCGATCTTATCTACGGGGACTTATCAGCCAACC |
| MarM | AATGATACGGCGACCACCGAGATCTACACAGTCAAACACTCTTTCCCTACACGACGCTCTTCCGATCTCGGGGACTTATCAGCCAACC |
| MarN | AATGATACGGCGACCACCGAGATCTACACAGTTCCACACTCTTTCCCTACACGACGCTCTTCCGATCTTCGGGGACTTATCAGCCAACC |
| MarO | AATGATACGGCGACCACCGAGATCTACACATGTCAACACTCTTTCCCTACACGACGCTCTTCCGATCTGATACGGGGACTTATCAGCCAACC |
| MarP | AATGATACGGCGACCACCGAGATCTACACCCGTCCACACTCTTTCCCTACACGACGCTCTTCCGATCTTATCTACGGGGACTTATCAGCCAACC |

file, TA sites in regions where two gene annotations overlap are presented for both of the overlapping genes. This method allows us to obtain gene essentiality calls for each condition, including the Inoculum, separately. Genes that were essential in the Inoculum but not during later time points were considered an artifact and were subsequently accounted for by limiting the essential core to the intersection of essential genes among all conditions. Furthermore, one gene was removed, due to a lack of MMAR ID. The essential core from this study was compared with genes being identified as essential in three other studies. Only orthologous genes were considered (E11 orthologs given by Weerdenburg et al. (45), H37Rv orthologs given by mycobrowser.epfl.ch, release 3, 5 June 2018) and only *M. marinum* M strain genes with a MMAR ID were used. This was done using R 4.2.1.

For conditional essentiality analysis, the resampling method of TRANSIT was used with the mean over non-zero sites (nzMean), effectively excluding low-permissive sites. Prior to that, the samples were normalized to contain $10^6$ reads per sample. The TRANSIT resampling method allows us to compare a test condition to a reference condition by subtracting the mean read count in the test condition from the mean read count in the reference condition, to obtain a difference of means. A null distribution was calculated by permutating genes among sites and samples ("pooled" option) and used to obtain an associated *P*-value. The corresponding TRANSIT output is documented in Table S3.

## *M. marinum* cluster analysis for the evolution of the GD/GA scores

The gene clusters analysis was performed with the R programming language, using the content of Table S2 as input. Genes with at least 5 TA sites and a normalized read count >1 in any of the samples were selected. For each gene, we computed its profile across time points by considering the average of the normalized gene count for each condition. The Manhattan distance between each pair of gene profile was computed and a hierarchical clustering algorithm with ward.D linkage was run. The resulting clustering tree was cut at a height of 35, allowing us to retain nine clusters (Table S3).

## Gene ontology analysis

Before functional group analysis via KEGG and GO enrichment, genes were thresholded as follows: fitness advantage (FA), $\log_2 fc \geq 0.585$, fitness disadvantage (FD), $\log_2 fc \leq$

−0.585, FA and FD, abs(log$_2$fc) ≥0.585. Genes were additionally thresholded by *P*-value as follows: 24 hpi and 48 hpi, *P*-value ≤ 0.1, 2 × 48 hpi, *P*-value ≤ 0.01. Different significance thresholding was motivated by better visualization of enriched terms.

Prior to submission to the enricher function or enrichKEGG function of the R package clusterProfiler, only genes with a MMAR ID were used and mapped to UniProt AC using the mycobrowser annotation (release 3, 28 March 2018) and the UniProt annotation (queried for "*Mycobacterium marinum*") (92). GO and KEGG term analyses were done with R (version 4.2.1) and the package clusterProfiler (version 4.4.4, 10.1016/j.xinn.2021.100141), respectively. For KEGG analysis, the KEGG annotation (release 103.1, 1 September 2022) (93) version was used, for GO terms, the QuickGO annotation for *Mycobacterium marinum* (version from 19.03.2021) was used (94). GO term names were extracted using the R package GO.db version 3.15.0. Group size was limited between 2 and 200 genes and only filtered by *P*-value < 0.01 for GO term analysis and *P*-value < 0.05 for KEGG analysis. For GO term analysis, all three aspects of the ontology were used: biological processes, molecular function, and cellular component.

## STRING database analysis

The analysis was performed using Cytoscape 3.9.1 (95) with Omics Visualizer 1.3.0 (58) and the STRING database version 11.5 (96). First, log$_2$fc and *P*-values were taken by comparing 24, 48, and 2 × 48 hpi to the inoculum via the TRANSIT resampling methods, as described above. Genes were filtered for absolute log$_2$fc ≥ 0.585, *P*-value ≤ 0.05 in any of the three time points, and converted to each respective ortholog in *Mycobacterium tuberculosis*, for which the full STRING network was subsequently retrieved. The network was cut at a confidence cutoff of 0.8 and singletons were omitted. The color scale in the layered rings represents from inside to outside the log$_2$fc of 24, 48, and 2 × 48 hpi, respectively, compared to the inoculum and was chosen to be capped at an absolute value of 4, for better visibility of colors.

## ACKNOWLEDGMENTS

We acknowledge the staff from the core facilities IGE3 Genomics platform at the Faculty of Medicine for their precious help. We thank Dr. H. Koliwer-Brandl and Dr. Cristina Boehm-Bosmani for their involvement in various preliminary experiments.

This work was supported by multiple grants from the National Centre for the Replacement, Refinement and Reduction of Animals in Research (NC3Rs) (awarded to G.R.S. and T.S.), the SystemsX.ch grant HostPathX to T.S., and the Swiss National Science Foundation research grants 310030_169386 and 310030_188813 to T.S.

## AUTHOR AFFILIATIONS

[1]Department of Biochemistry, Faculty of Science, University of Geneva, Science II, Geneva, Switzerland
[2]Department of Microbial Sciences, School of Biosciences, University of Surrey, Guildford, Surrey, United Kingdom
[3]Department of Microbiology and Molecular Medicine, Faculty of Medicine/CMU, University of Geneva, Institute of Genetics and Genomics in Geneva (iGE3), Genève, Switzerland
[4]Bioinformatics Support Platform for data analysis, Geneva University, Medicine Faculty, Geneva, Switzerland

## AUTHOR ORCIDs

Louise H. Lefrançois  http://orcid.org/0000-0002-9544-8435
Jahn Nitschke  http://orcid.org/0000-0003-3838-0945
Gaël Panis  http://orcid.org/0000-0002-6926-6224
Julien Prados  http://orcid.org/0000-0002-8546-241X

Rachel E. Butler ⬤ http://orcid.org/0000-0002-6018-7765
Tom A. Mendum ⬤ http://orcid.org/0000-0002-6331-2605
Nabil Hanna ⬤ http://orcid.org/0000-0002-3288-5486
Graham R. Stewart ⬤ http://orcid.org/0000-0002-6867-6248
Thierry Soldati ⬤ http://orcid.org/0000-0002-2056-7931

## FUNDING

| Funder | Grant(s) | Author(s) |
| --- | --- | --- |
| National Centre for the Replacement Refinement and Reduction of Animals in Research (NC3Rs) | NC/M002012/1 | Thierry Soldati |
| | | Louise H. Lefrançois |
| | | Huihai Wu |
| | | Rachel E. Butler |
| | | Thomas A. Mendum |
| | | Graham R. Stewart |
| Schweizerischer Nationalfonds zur Förderung der Wissenschaftlichen Forschung (SNF) | 310030_169386 | Thierry Soldati |
| | | Louise H. Lefrançois |
| | | Jahn Nitschke |
| | | Nabil Hanna |
| Schweizerischer Nationalfonds zur Förderung der Wissenschaftlichen Forschung (SNF) | 310030_188813 | Thierry Soldati |
| | | Louise H. Lefrançois |
| | | Jahn Nitschke |
| | | Nabil Hanna |
| SystemsX.ch (Swiss SystemsX.ch) | HostPathX | Thierry Soldati |
| | | Louise H. Lefrançois |
| | | Nabil Hanna |

## AUTHOR CONTRIBUTIONS

Louise H. Lefrançois, Conceptualization, Data curation, Formal analysis, Investigation, Methodology, Validation, Visualization, Writing – original draft, Writing – review and editing | Jahn Nitschke, Conceptualization, Data curation, Formal analysis, Investigation, Software, Validation, Visualization, Writing – original draft, Writing – review and editing | Huihai Wu, Data curation, Formal analysis, Software, Visualization, Writing – original draft | Gaël Panis, Data curation, Formal analysis, Visualization, Writing – original draft, Writing – review and editing | Julien Prados, Data curation, Formal analysis, Software, Visualization, Writing – original draft, Writing – review and editing | Rachel E. Butler, Conceptualization, Investigation, Methodology, Writing – review and editing | Tom A. Mendum, Conceptualization, Investigation, Methodology, Writing – original draft | Nabil Hanna, Conceptualization, Data curation, Formal analysis, Investigation, Visualization, Writing – original draft, Writing – review and editing | Graham R. Stewart, Conceptualization, Funding acquisition, Investigation, Methodology, Project administration, Supervision, Writing – original draft, Writing – review and editing | Thierry Soldati, Conceptualization, Funding acquisition, Project administration, Supervision, Writing – original draft, Writing – review and editing

## DATA AVAILABILITY

The HiSeq data sets are available in the Sequence Read Archive (SRA) repository, Bioproject accession number PRJNA967609. The supplemental tables and files are available in the Gene Expression Omnibus (GEO) repository, accession number GSE249096.

## ADDITIONAL FILES

The following material is available online.

### Supplemental Material

**Legends (mSystems01326-23-S0001.docx).** Supplemental figure legends.
**Figure S1 (mSystems01326-23-S0002.pdf).** PCA of all experimental conditions in *D. discoideum* and library saturation.
**Figure S2 (mSystems01326-23-S0003.pdf).** Differential representation of *M. marinum* mutants
**Figure S3 (mSystems01326-23-S0004.pdf).** Hierarchical clustering of log2 fold changes in *D. discoideum*.
**Figure S4 (mSystems01326-23-S0005.pdf).** Fitness advantage and fitness disadvantage over the infection time course in *D. discoideum*.
**Figure S5 (mSystems01326-23-S0006.pdf).** Clustering of differentially affected genes of *M. marinum* during infection of BV2 microglial cells.
**Table S2 (mSystems01326-23-s0007.xlsx).** Gene counts for *D. discoideum* and BV2 experiments.
**Table S3 (mSystems01326-23-s0008.xlsx).** Conditional essentiality analysis for *D. discoideum* and BV2 experiments.
**Table S4 (mSystems01326-23-s0009.xlsx).** Enrichment analyses for *D. discoideu*m and BV2 experiments.

### Open Peer Review

**PEER REVIEW HISTORY (review-history.pdf).** An accounting of the reviewer comments and feedback.

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
