## [Reviewer comments · mSystems]

Temporal genome-wide fitness analysis of *Mycobacterium marinum* during infection reveals the genetic requirement for virulence and survival in amoebae and microglial cells

Louise Lefrançois, Jahn Nitschke, huihai Wu, Gaël Panis, Julien Prados, Rachel Butler, Thomas Mendum, Nabil Hanna, Graham Stewart, and Thierry Soldati

Corresponding Author(s): Thierry Soldati, Universite de Geneve

Review Timeline:

Submission Date:

December 15, 2023

Accepted:

December 15, 2023

Editor: Jack Gilbert

Reviewer(s): The reviewers have opted to remain anonymous.

Transaction Report:

DOI: <https://doi.org/10.1128/msystems.01326-23>

Re: mSystems01326-23 (Temporal genome-wide fitness analysis of *Mycobacterium marinum* during infection reveals the genetic requirement for virulence and survival in amoebae and microglial cells)

Dear Prof. Thierry Soldati:

Your manuscript has been accepted, and I am forwarding it to the ASM production staff for publication. Your paper will first be checked to make sure all elements meet the technical requirements. ASM staff will contact you if anything needs to be revised before copyediting and production can begin. Otherwise, you will be notified when your proofs are ready to be viewed.

Featured Image Submissions: If you would like to submit a potential Featured Image, please email a file and a short legend to mSystems@asmusa.org. Please note that we can only consider images that (i) the authors created or own and (ii) have not been previously published. By submitting, you agree that the image can be used under the same terms as the published article. File requirements: square dimensions (4" x 4"), 300 dpi resolution, RGB colorspace, TIF file format.

Sincerely,
Jack Gilbert
Editor
mSystems